# Support needs of south Asian adult survivors of childhood sexual abuse in the UK: Perspectives of UK mental health professionals and key stakeholders

Yuqian Chen[1], Eugenia Drini[1], Rebecca Appleton[2], Jo Billings[1], Shivangi Talwar[1,3]*

**1** Division of Psychiatry, University College London, London, United Kingdom, **2** NIHR Mental Health Policy Research Unit, University College London, London, United Kingdom, **3** Department of Psychology, Kingston University, London, United Kingdom

* shivangi.talwar.21@ucl.ac.uk

## Abstract

Childhood sexual abuse (CSA) can have persistent emotional and behavioural consequences in the lives of adult survivors. In the UK, South Asian CSA survivor may face distinctive cultural and structural barriers to care. This study aimed to explore the perspectives of UK mental health professionals and key stakeholders on the treatment and support needs of South Asian adult CSA survivors. We conducted semi-structured, in-depth interviews using interview guides with mental health professionals and key stakeholders working with adult CSA survivors and other trauma survivors of South Asian origin in the UK. We used reflexive thematic analysis to analyse data with the aid of NVivo software. We interviewed seven participants, including five mental health professionals and two stakeholders in the UK. We conceptualised our findings as a bilateral relationship between barriers to seeking help and the need to improve support. The barriers are an interplay of practical concerns around the background of professionals, difficulties supporting survivors, language discrepancies and specific cultural notions. There is a potential need to enhance psychosocial and informal support for the survivors in the UK. Help from families and within the community, as well as accessible, culturally tailored information for the community members could help. Improving the mental health literacy among the community and a better understanding of cultural needs by the service providers is a way forward. This research underscored the potential need for culturally tailored interventions and increased mental health literacy within the South Asian community and provided valuable insights for enhancing service delivery to this underrepresented population. Future research could incorporate insights of service users themselves, as well as include participants from more diverse demographic backgrounds.

**Data availability statement:** The interviews with mental health professionals and key stakeholders include information about their clients. Public availability is likely to compromise their clients' privacy. Therefore, the data cannot be made freely available. This is in line with the requirements of the Research Ethics Committee of University College London. Data can be made available upon reasonable request to me or the UCL Research Ethics Committee: ethics@ucl.ac.uk.

**Funding:** The authors received no specific funding for this work.

**Competing interests:** The authors have declared that no competing interests exist.

## Introduction

Experiencing trauma is common with approximately 70% of the population globally being exposed to a traumatic event at least once in their lives [1]. Whilst the exposure does not always translate into mental ill-health, it may lead to potential mental health problems such as Post-Traumatic Stress Disorder (PTSD), anxiety, psychosis and depression [2]. One of the most prevalent and debilitating developmental traumas, childhood sexual abuse (CSA), defined as sexual activity involving a child who is incapable of giving consent, enacted through power, coercion, or exploitation [3,4]. Adults with such intrusive experiences of CSA, often repeated and protracted, are more likely to be diagnosed with Complex Post-Traumatic Stress Disorder (CPTSD) [5]. These survivors, during their childhood and/or adult lives, may also experience co-morbid mental health and substance use difficulties [6], highlighting their complex treatment and support needs. In the UK, the Crime Survey for England and Wales estimated that 7.5% of adults aged 18–74 years experienced CSA before age 16, underscoring the public-health significance of tailored support [7].

The first line of treatment for PTSD includes Trauma-focused Cognitive Behavioral Therapy (TF-CBT), Eye Movement Desensitization and Reprocessing (EMDR), Narrative Exposure Therapy (NET), and Prolonged Exposure Therapy (National Institute for Health and Care Excellence, 2018). The evidence for treating CPTSD is still under development, but the NICE guidelines suggest catering to complex needs and increasing the length of the above treatments [NICE, 2018].

South Asians residing in the UK face unique community-specific challenges in accessing mental health support including stigma around mental health services, victim-blaming norms, and fear of bringing dishonour to their families [8–10]. Other barriers to accessing support by South Asians include language differences, distrust of healthcare providers, and limited availability of culturally sensitive models in clinical practice [11–13]. Mental health services across South Asia are relatively under-resourced, and individuals sometimes attribute psychological distress to supernatural causes, seeking support from religious leaders and traditional healers, due to their accessibility, affordability, and social acceptability [14,15]. As compared to adult survivors of CSA in Western societies, where individualistic cultures allow personal reflection and open expression, adult CSA survivors of South Asian origin are influenced by concepts such as *izzat* (honour), *sharam* (shame), and the pervasive culture of silence, experiencing further barriers in accessing support [16,17].

One way to understand the challenges faced by South Asian CSA survivors in accessing mental health support is through Campinha-Bacote's model of cultural competence. This model outlines five interrelated aspects: cultural awareness, cultural knowledge, cultural skill, cultural encounters, and cultural desire [18]. For example, the failure to address stigma or provide language-accessible services may reflect gaps in providers' cultural knowledge and skills. Limited understanding of culture-specific values may reduce opportunities for meaningful cultural encounters.

The National Healthcare Service (NHS) in the UK offers various services designed to assist trauma survivors, including assessment, active monitoring, psychological

therapies, and medication [19]. Alongside the NHS, numerous charities and organisations in the UK offer helplines, safe spaces, and counselling sessions for trauma survivors [20]. Cumulatively, these services have been shown to be disparate in terms of accessibility and effectiveness of the treatments offered, and particularly inequitable for ethnically underrepresented and economically disadvantaged communities [21]. South Asians form the largest ethnic minority in the UK [22]. Therefore, it is imperative that services address the challenges faced by the South Asian CSA survivors seeking help in the UK.

Addressing underrepresented populations in trauma research is gaining momentum [McClendon et al., 2022], but there remains a notable lack of evidence on the experiences of the South Asian community globally. South Asian community's distinct cultural norms, stigma surrounding mental health, and limited engagement with support systems can significantly shape survivors' trauma recovery pathways [23]. Therefore, filling such research gaps is essential in informing the development of culturally responsive interventions for South Asian CSA survivors. To date, commonly reported themes in the limited literature include notions of honour, shame and modesty, which impact help-seeking behaviours among South Asian women [24–26]. These studies explore the unique coping strategies developed by South Asian trauma survivors, including transforming cultural silence into a space for healing, accessing community support networks, practicing cultural healing modalities, engaging in social advocacy, and developing intentional self-care practices [25,26]. Studies also call for culturally-informed interventions, given associations found between CSA experiences and adverse mental health outcomes [24].

A recent systematic review of 24 studies synthesised the impact of CSA on adult survivors in South Asian countries, and treatments and support offered to them [27]. The review strikingly demonstrated that evidence was available from only four of the eight South Asian countries, predominantly from India. The review found South Asian survivors reported mental health and interpersonal relationship difficulties and unpleasant responses to disclosure of abuse [27]. A further finding from the review was the absence of an evidence base for specific treatments, with CBT being the most reported intervention. In Pakistan, clinical reports shared some successful treatments in trauma care for female CSA survivors in Pakistan, emphasizing the need for establishing safety and unconditional acceptance of health professionals [28]; while policy commentary from Nepal appealed a response to Violence Against Women and Girls in Nepal, urging for policy changes to protect women and girls being abused [29]. Whilst the research with adult CSA survivors is still in its infancy in South Asia, the needs of survivors from this ethnic group are gradually receiving increased recognition.

In the UK context, in-depth interviews with British South Asian women revealed that cultural stigma, familial shame, and fear of community ostracism significantly impeded participants from reporting sexual violence and childhood sexual abuse [17]. In the USA, a qualitative study with immigrant South Asian women survivors of CSA proposed a dynamic model suggesting that the South Asian cultural contexts of CSA survivors could influence both the understanding of the abusive experience, and the salience and selection of resilience strategies [26].

Another study with South Asian adult CSA survivors in the USA proposed the Idealized Cultural Identities Model. This model suggested that ethnic minority groups create idealized identities, using cultural stereotypes and dominant group expectations to navigate social marginality, impacting their help-seeking behaviours and perceptions of issues such as CSA [25]. Supporting this theoretical framework in the USA, a cross-sectional survey found significant associations between CSA and both relationship violence and suicide attempts among South Asian survivors [24].

A comprehensive understanding of the interplay between cultural context and trauma response is imperative to ensure that service design and interventions employed are appropriate and effective [30]. The intrinsic impact of CSA on adult survivors and the South Asian cultural norms complicate the disclosure of abuse experiences [17,26]. However, utilising professionals' views on the needs of and treatments for people requiring support could shed light on the needs and preferences of service users [31,32]. Further, specifically for adult CSA survivors, mental health and non-statutory support could both be instrumental in the survivors' recovery. Therefore, this study aimed to explore the perspectives of UK mental health professionals and key stakeholders on the treatment and support needs of South Asian adult CSA survivors.

Specifically, this study examined: barriers to help-seeking and engagement; perceived gaps in current service provision; and support needs and practical improvements to care.

## Methodology

### Ethics statement

This study received ethical approval from the UCL Research Ethics Committee (Ethics ID number 22535/002). We followed the Reflexive Thematic Analysis Reporting Guidelines [33] for reporting the study.

### Participants

We included mental health professionals and key stakeholders who had experience of working with adult South Asian clients affected by trauma, specifically CSA, within the UK. There was no upper age limit, but the lower age limit was 23 years for mental health professionals to ensure they had completed appropriate professional qualifications, and 18 years for other key stakeholders to ensure eligibility to work in the UK. Inclusion criteria for mental health professionals required participants to be qualified practitioners with at least two years of experience working with CSA survivors of South Asian origin. For key stakeholders, defined as non-clinical roles with direct, substantive experience supporting this population (e.g., leaders or researchers within charities serving South Asian communities), needed to have direct experience supporting South Asian CSA survivors to be included in this study. No survivors took part in this study.

### Recruitment

We used purposive and snowball sampling to recruit participants. Using maximum variation sampling, we aimed for participants from diverse ethnic backgrounds, genders and varying levels of work experience. We invited professionals to take part in our study by emailing professionals in the network of JB (consultant clinical psychologist in the UK) and ST (clinical psychologist and researcher specialising in working with South Asian CSA survivors). We also posted our recruitment flyer on LinkedIn and Twitter. Additionally, we informed and invited relevant UK organisations working with adult CSA survivors and South Asian communities. Those who took part were requested to pass on the flyer to their colleagues. Potential participants were provided with an information sheet about the study. Written consent was obtained from all participants who agreed to take part.

### Data collection

YC conducted semi-structured in-depth interviews, either face-to-face or online via MS Teams, depending on participants' availability and preference. The interview guide for this study was adapted from a guide developed by ST, RA and JB, and previously used in a related qualitative study led by ST with mental health professionals and key stakeholders in South Asia [27], YC and ED further revised and contextualised the interview guide for UK services. YC, after conducting the first two interviews, edited the topic guide further in response to the completed interviews, and in consultation with ST. Before interviews, participants were informed about the sensitive nature of CSA discussions; they were then reminded of their right to pause, skip, or stop the conversation without consequences. We also included details on data collection, storage and contact means of researchers in our information sheet. The interviews began with general questions on participants' roles and their interactions with clients affected by trauma. Subsequently, participants were asked about their views and approaches to offering support to adult survivors of CSA of South Asian clients (see Supplementary Materials S1 File for interview guide used with mental health professionals and S2 File for interview guide used with stakeholders). All interviews were audio-recorded with participants' consent and were transcribed by YC for analysis. ST reviewed two transcripts to advice YC on the interview process and ensure that the interview was in line with the study objectives.

## Data analysis

We analysed the transcripts using reflexive thematic analysis [33]. We chose to employ this model to draw on our professional and personal reflexivity to identify patterns, from a critical realist orientation [33]. The first author, YC, familiarised herself with the data by transcribing the interviews and reading and re-reading the transcripts. She further recorded brief reflexive notes to develop familiarity with the data. Then, YC inductively coded the data using NVivo 14 (14.24.0) to address how mental health professionals and stakeholders understand and respond to the needs of South Asian CSA survivors. These codes were discussed with ST to develop candidate themes reflecting meaningful patterns across the data. The potential themes were then reviewed through discussion by describing the developed themes and subthemes to identify the relationship between them. The labels were refined by YC, ED and ST. All team members reviewed the resulting analysis, and further changes were made to the developed themes, sub themes and their labels.

## Quality and rigor

This study employed several strategies to ensure quality and rigor. Credibility was enhanced through an interpretative reflexive process [34], where one person led data collection, while two additional team members actively supervised the analysis. Wider team members were involved in manuscript drafting. The first author maintained a reflexive journal to document decision-making, and reflections throughout the study [35].

The study aimed to recruit participants from diverse backgrounds, including ethnicity, gender, age, and occupation, to capture a range of opinions. We collected data through both online and in-person meetings, balancing the participant-researcher relationship with convenience and accessibility, thereby increasing the likelihood of participation. Consistent with reflexive thematic analysis, we did not calculate inter-rater reliability; instead, we enhanced credibility via audit trail and repeated supervision meetings.

## Reflexivity

All team members have academic mental health backgrounds. The first author, YC, is a female postgraduate student from China studying in the UK. Coming from an East Asian background sharing some cultural commonalities with the studied population, she found some of the cultural nuances and client experiences relatable. This shared cultural background also led at times to empathising with the challenges faced by the South Asians in the UK. At the same time, YC felt distant in forming a complete understanding of South Asian and UK cultures throughout the study. YC was closely supervised by ST and ED, qualitative researchers and clinical academics.

ED is a Greek female counselling psychologist and an academic in the UK with experience of working in the NHS and conducting qualitative research. She has been working with people affected by trauma and diverse populations and is interested in their experiences of treatment, and their individual needs.

JB is a consultant clinical psychologist and clinical academic, with over 20 years of experience of working in the NHS and specialist UK trauma services. She is also an academic with extensive experience of conducting qualitative research with traumatised populations and professionals working to support them.

RA is a senior researcher with experience and interest in working with children and young people with mental health conditions. She has extensive experience and interest in conducting policy reviews, qualitative research and implementation science.

ST is an Indian female clinical psychologist qualified in India with experience of undertaking qualitative research in South Asia and the UK, especially with adult survivors of CSA. ST, JB and RA have previously explored views of similar professionals working with adult CSA survivors in South Asia [27].

## Results

Seven participants took part in this study, including five mental health professionals and two stakeholders. Six participants identified as female, and one as male. In terms of ethnicity, one participant was Black British, two were British Indian, and four participants were White (British, German and Irish). The age of our participants ranged between 30–66 years, with a mean age of 47 years. Table 1 describes further details of the characteristics of our participants.

The interviews lasted between 30 minutes 26 seconds and 59 minutes 53 seconds, with an average duration of 39 minutes 40 seconds.

We have conceptualized the findings as a bidirectional relationship between barriers to professional help and the need to enhance support systems. Within these two domains we developed four themes: 1. difficulties accessing support, 2. challenges faced by survivors with providers, 3. informal support and 4. improving support. Fig 1 shows an illustration of the themes and sub-themes throughout the study. A detailed breakdown of these themes and sub-themes is presented in Table 2.

### Barriers to professional help

Two themes were included within this domain explaining the multiple challenges that professionals think South Asian adult trauma survivors, specifically CSA survivors, in the UK face in accessing mental health services. These themes are: 1. Difficulties accessing support and, 2. Challenges faced by survivors with professionals.

### Difficulties accessing support

Our participants highlighted that their South Asian clients are likely to encounter several obstacles when deciding to seek help regarding their trauma, specifically CSA. We have explained these below in two sub-themes: 1.1 Pragmatic concerns and 1.2 Culture-specific factors.

**Pragmatic concerns.** Despite NHS mental health services being free to recipients at the point of access, the participants in our study reported that their South Asian clients sometimes face barriers to accessing

**Table 1. Characteristics of participants.**

| Category | Subcategory | N(%) |
|---|---|---|
| Occupation | Stakeholders | 2 (28.58) |
| | *Researcher in a charity* | *1 (14.29)* |
| | *Charity chair* | *1 (14.29)* |
| | Mental health professionals | 5 (71.42) |
| | *Clinical Psychologist* | *3 (42.85)* |
| | *Psychotherapist* | *2 (28.57)* |
| Sex | Female | 6 (85.71) |
| | Male | 1 (14.29) |
| Age | 20-39 | 3 (42.86) |
| | 40-59 | 2 (28.57) |
| | 60 | 2 (28.57) |
| Ethnicity | Black British | 1 (14.29) |
| | British Indian | 2 (28.57) |
| | White | 4 (57.14) |
| | *British* | *2 (28.57)* |
| | *German* | *1 (14.29)* |
| | *Irish* | *1 (14.29)* |

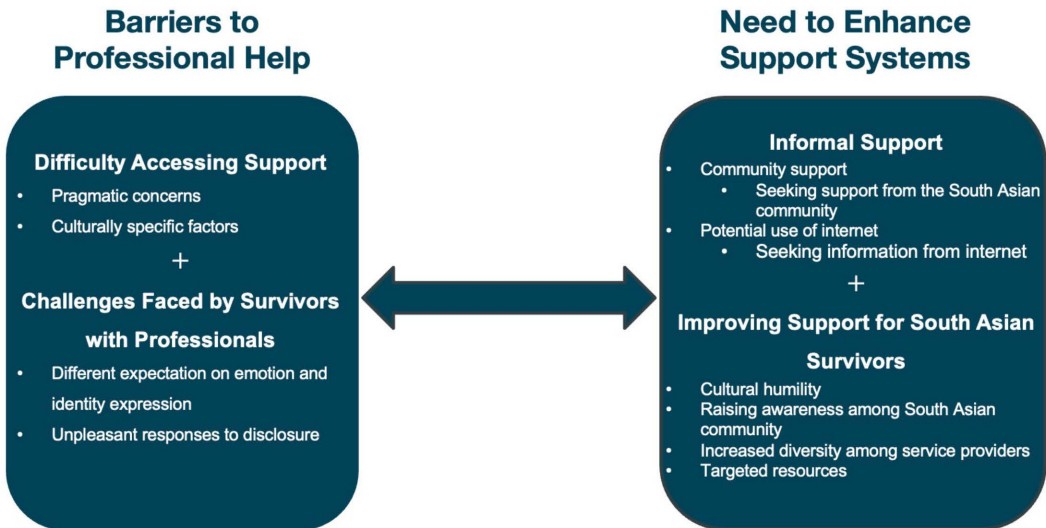

**Fig 1. Overarching themes, themes and sub-themes.**

comprehensive support due to **financial costs**. They suggested that long NHS waiting lists could drive clients to consider private practitioners who can offer immediate support at a higher expense. The mental health professionals highlighted that many survivors may lack sufficient financial means to afford long-term treatment. Beyond therapy costs, mental health professionals noted that clients also face other costs of accessing help such as transportation expenses.

*And then it's of course connected to quite some financial cost and then, getting successful support but it being so expensive that they couldn't afford it anymore.*

*- Academic and Clinical Psychologist, NHS*

A stakeholder explained that **language barriers** further complicated access to suitable help. Not all service providers were equipped in the diverse languages spoken within the South Asian community. Specifically for stakeholders working in charities, the feasibility of using professional interpreters was impacted by their charges and accessibility.

*We have people who have English as a foreign language and have to fight the difficulty to communicate.*

*- Key stakeholder and Charity chair*

A mental health professional spoke about the **limited population of South Asian representatives** in mental health or charity organisations. According to them, this lack of diversity in the ethnic backgrounds of professionals may reduce the appeal of accessing professional help.

*I know some people from different backgrounds have requested therapists from those backgrounds, and we often can't offer that. Most of our therapists are white, female, British […]*

*- Academic and Clinical psychologist, NHS*

**Table 2. Themes and sub-themes.**

| Barriers to Professional Help | | | Need to Enhance Support Systems | | |
|---|---|---|---|---|---|
| Difficulties accessing support | Pragmatic concerns | Financial costs | Informal Social Support | Seeking support from the South Asian community | Strong family bonds |
| | | Language barriers | | | |
| | | Limited population of South Asian representatives | | | |
| | | Difficulties finding communities | | | |
| | Culture-specific factors | Extra level of shame | | Seeking information from the internet | Discreet and accessible information and support |
| | | | | Concern about confidentiality | |
| | | | | Strong community and family influence | |
| | | | | Handle these issues on their own without professional help | |
| | | | | Delayed disclosure | |
| | | | | Faith crisis | |
| Challenges faced by survivors with professionals | Different expectations on emotion and identity expression | Reserved vs. Open expression | Improving support for South Asian CSA survivors | Cultural humility | Open to the unknown |
| | | | | | Appropriate adjustments |
| | | | | Raising awareness among South Asian community | psychoeducation |
| | Unpleasant responses to disclosure | Not being believed | | Increased diversity among service providers | Increased representation of cultural identities |
| | | | | Targeted resources | Documents for clients in various languages |
| | | | Facilities avoid getting involved | | Disseminating relevant information in venues and events relevant to South Asian culture |
| | | | | | adapt Western-based interventions and assessments |

Beyond professional support, a mental health professional reported pragmatic concerns in seeking informal support. They mentioned that clients from certain South Asian countries might face **difficulties finding communities** that share their cultural background. The professional explained that for clients from countries with large population based in the UK, such as Pakistan or Afghanistan, there is likely to be better availability of resources including shops selling food from their home countries to remind clients of their roots. They suggested that such networks can provide a sense of cultural safety and understanding that is particularly important when dealing with sensitive issues like CSA. However, according to the professional, for their clients from cultures with smaller populations, experiencing a sense of belonging and security can be more challenging.

*I think countries like Nepal, for instance, it's much harder to find either that particular group of people or even that particular network and that sense of safety around it.*

*- Clinical psychologist, NHS*

**Culture-specific factors.** Participants reported multiple culture-specific factors that affected the decision-making of South Asian CSA survivors in seeking professional support. One of the most significant factors reported was the **extra level of shame** attached to CSA for individuals from South Asian backgrounds. Mental health professionals and stakeholders reported that survivors of any ethnic background would share similar concerns around CSA: embarrassment, self-blame, and guilt. As stated by our participants, South Asian CSA survivors appear to experience an additional layer of shame on top of that reported by survivors from other ethnicities. According to the professionals, this added-on difficulty could be attributed to the lack of discussions on CSA as being abusive. Our participants highlighted that this sense of shame was further compounded by the fear of damaging the family's honour.

*I think the extra layer is the fear of reprisals, the fear of what will people think of you, the fear of ruining the family name.*

*- Psychotherapist. National mental health association*

In our study, mental health professionals reported that South Asian clients may struggle with trusting clinicians leading to **concerns about confidentiality**. They highlighted that where some South Asian clients may not express themselves fluently in English, interpreters' involvement would be reasonable yet sometimes disturbing to the clients. According to our participants, some clients might feel more reluctant to disclose in the presence of an interpreter. The participants said that their South Asian clients would be residing in the UK but may fear that their information would be leaked into the community in the UK or their home countries.

*So I think there's often a worry that, it might come out into their communities in some way or another, specifically, when working with an interpreter, people are really worried that the information gets carried outside of the room.*

*- Clinical psychologist, NHS*

Mental health professionals and stakeholders in our study also addressed the **strong community and family influence**. They said that this could include the dowry system, where being sexually abused would impact the financial worth of the family name. Our participants highlighted intra-familial abuse among South Asian communities, leading to hesitation among clients to report abuse within their families. They said that an additional burden of respecting the older generation in this ethnic group would forcefully induce clients' silence.

*If you say something so horrendous, the whole family name will be tarred. The older generation will do anything in their power to protect the family name. From outside, it's gotta look good, they are lovely, respectable families. And if you come along and say no, actually I was abused, that's gonna completely shatter the illusion that they were a respectable family.*

*- Psychotherapist, National mental health association*

In our study, mental health professionals and stakeholders pointed out that South Asian communities also seem to have limited information on mental health and CSA, leading to increased stigma toward CSA survivors, as well as decreased help-seeking behaviours. Participants worried that with the unawareness of the unlawful nature of CSA, survivors were less likely to recognise the need to seek support. Participants also spoke about the lack of importance placed on CSA incidents within South Asian communities. According to the professionals in our study, this could lead survivors to believe they should **handle these issues on their own without professional help**. A mental health professional mentioned that South Asian clients, as compared to their White British counterparts, often felt less compelled to seek assistance and were sometimes less aware of resources available to them.

*And I think maybe sometimes the level of importance that's attached to the charge of sexual abuse, [...] it's often like, almost has less importance because it doesn't get spoken about and therefore it's normalized in a weird backward way.*

*- Clinical psychologist, NHS*

Another distinct presentation observed by mental health professionals and stakeholders in South Asian CSA survivors was **delayed disclosure**. Participants proposed that compared to other clients, their South Asian clients could take longer to comprehend their experiences as abusive and disclose their experiences.

*I think there's an element of tolerating it a lot more and longer because of fear of reprisals from family and community, [...] I think in the South Asian community it's reported less.*

*- Psychotherapist, National mental health association*

Participants reflected upon delayed disclosure by their South Asian survivor clients as a consequence of clients' tendency to suppress their emotions. They described how survivors often believe that their emotions will not be acknowledged or validated.

*If you've been told all your life this is your fault this happened to you or this shouldn't have happened if you were good, […] so you can't have those feelings of upset.*

*- Academic and clinical psychologist, NHS*

A mental health professional discussed this particularly in male clients, who may find it challenging to articulate their experiences. Further, they reported that South Asian male clients often had a limited understanding of their emotional responses and struggled to connect with or express their feelings, which could further complicate their treatments.

*I mean what I find with male clients from Southeast Asian backgrounds actually would be that there's very limited understanding or space for any sort of emotional response to things... and I guess that also applies to the child sexual abuse that's always seen as like something that happened and it gets spoken about very pragmatically. But if you'd ask them to kind of delve into that with a little bit more detail, that would be really difficult...*

*- Clinical Psychologist, NHS*

Another cultural factor mentioned by mental health professionals and stakeholders was regarding the religious base in some South Asian communities. According to our participants, despite being occasionally a protective factor and providing clients with hope and survival, religion could also potentially lead to a **faith crisis.** This could be damaging for CSA survivors, with them questioning:

*Is it something you deserve? Is God punishing you? Is it something that you did? Is it your fault that you're dressing/ acting too provocatively? I know we have that in other societies as well, but it's just been quite pronounced that you must have done something wrong if you were assaulted as a child; you must have asked for that in a way; otherwise, something this bad wouldn't happen to a child.*

*- Academic and clinical psychologist, NHS*

Some mental health professionals noted that some South Asian clients may have a different understanding of mental health due to cultural norms, particularly the role of spirituality, which could influence their motivation to seek professional help.

*I think particularly for South Asian clients the role of sort of spirituality or evil eye can mean that there are alternative narratives for what is manifesting in terms of trauma symptoms or mental health symptoms.*

*- Clinical psychologist, NHS*

One mental health professional in this study believed that the historical background of the South Asian population in the UK could lead to the excessive need to compete against each other. This could be related to surviving in a foreign country for resources, therefore denying themselves the opportunity to work on their psychological difficulties.

*So, then what happens is there is a massive drive to compete with each other to show everybody that I'm still surviving in this Western landscape, despite my historical landscape. Now, this is all intergenerational trauma. It's all in our genes.*

*- Psychotherapist at a private practice*

### Challenges faced by survivors with professionals

In this study, participants reported that their South Asian adult CSA survivor clients, at times, faced difficulties during the interaction with the professionals. These challenges could be 2.1 different expectations by service users and service providers on emotion and identity expression and 2.2 unpleasant responses to disclosure.

**Different expectations on emotion and identity expression.** Mental health professionals in our study said that in the South Asian culture, overt emotional displays would be potentially considered disruptive and abnormal. Hence, our participants' clients were likely to prioritise restricted expression within and outside therapeutic settings. They said that this is in contrast to their clinical training in the UK where open expression of emotions and a focus on individuality were often considered the norms in therapeutic conversations. Whilst these service providers did not necessarily report interpreting the reticent nature of South Asian clients as resistance or lack of engagement, they said that they find it challenging during sessions. Additionally, our participants thought that identity expression by South Asian clients may be closely tied to family and community, leading to potential challenges in discussion regarding individual purposes during the recovery process.

Our participants reported that cultural differences in emotional expression also significantly impacted how survivors processed and attributed blame for their abuse experiences. Whilst Western therapeutic approaches often encourage survivors to externalize blame onto perpetrators, South Asian clients could struggle with this aspect. This could potentially reinforce survivors' tendencies toward self-blame.

*I think... that blame placed very strongly on the person and that self-blame isn't as it would be maybe in some of my white British clients more coming from internal, but quite explicitly external from sisters, brothers, family members, parents.*

*- Academic and clinical psychologist, NHS*

**Unpleasant responses to disclosure.** Mental health professionals and stakeholders in our study reported that South Asian service users who found the courage to disclose traumatic experiences often held a profound hope for understanding and validation. However, when they did reach out to family, community members or mental health professionals, these hopes frequently remained unfulfilled. According to these mental health professionals and stakeholders, **not being believed** turned out to be a common response received by South Asian CSA survivors when they attempted to disclose, exacerbating the self-blame and shame aspect of their experiences. Moreover, the professionals said that sexual abuse was an even less-believed act, given its covert nature compared to other forms of abuse that were

more easily witnessed. Participants pointed out that such contrast between expectation and reality could have long-term effects on survivors' confidence and self-esteem. It could reduce clients' willingness to seek professional help in the future and affect trust building with service providers in healthcare systems.

*So in the cases I know of, they have reached out [for help] before and haven't received the right help; [... They have] quite bad experiences of not being believed, or their feelings not being kind of validated and that they're making stuff up and that they're making a big deal out of things.*

*- Academic and Clinical psychologist, NHS*

A mental health professional also addressed the possibility that a staff member from a Western organisation may **avoid getting involved** in the treatment of South Asian clients due to the lack of knowledge of their culture. Such a response might reinforce feelings of isolation and helplessness among South Asian CSA survivors. It is likely to delegitimise the survivors' trauma and redirect them to feeling misunderstood or endangered.

*Staff often work quite disregarding and saying, ohh you do things differently, so you sort it in your own community, we're not gonna get involved. So often like turning a blind eye or not actually asking enough questions because it was a different culture. So, maybe having different standards.*

*- Academic and Clinical psychologist, NHS*

## Need to enhance support systems

We have organised two themes in this domain to emphasize the importance of improving both informal and formal support mechanisms for adult CSA survivors of South Asian origin. These two themes are 3. Informal social support and, 4. Improving support for South Asian CSA survivors.

### Informal social support

According to our participants, alongside or instead of seeking professional mental health support, South Asian CSA survivors may turn to informal sources of support. Based on our interviews with professionals, we developed two sub themes: 3.1 Seeking support from the South Asian community and 3.2 Seeking information from the internet.

**Seeking support from the South Asian community.** Given the strong inter-familial bond within South Asian culture, our participants said that family and community support could be very helpful for clients affected by CSA. Mental health professionals and stakeholders addressed the possibility of **strong family bonds** having either a positive or negative effect depending on the individual, family or community.

*If somebody just disclosed childhood sexual abuse or family members that people say ohh OK, we must make sure to protect you, we believe you, we will put everything into place so that you don't get reimposed to that person, or it can go the other way.*

*- Clinical psychologist, NHS*

Our participants voiced that there were numerous benefits to having family support. They said that their clients could gain emotional and psychological security from the positive feedback they receive from close family members. The professionals also said that their South Asian survivor clients could benefit from practical assistance, such as help with managing their schedules and attending therapy sessions. According to one mental health professional in this study, the South

Asian population living in the UK would have an opportunity to engage with people of other cultures. They said that these different social dynamics would allow for the opportunities to experience openness, acceptance, and support beyond traditional family systems.

**Seeking information from the internet.** One stakeholder suggested that the Internet could be a potential means for **discreet and accessible information and support**. They said that the anonymity of the internet could allow survivors to explore sensitive topics, as well as potential resources for support. The stakeholder further suggested that online platforms could connect individuals with similar experiences, fostering networks that could be supportive.

*And I think the Internet can be both a facilitator and a barrier [...] people can share their lived experience, their truth on social media.*

*- Key stakeholder and researcher at charity*

### Improving support for South Asian CSA survivors

According to mental health professionals in our study, the support provided was aligned with NICE guidelines, such as trauma-focused CBT, EMDR, compassion-focused therapy, and narrative exposure therapy. Of note, our participants highlighted that support was almost always offered regardless of cultural background, identity or type of trauma. Mental health professionals and key stakeholders also disclosed that they treated their South Asian clients in similar ways as compared to non-South Asian clients, with only nuanced adjustments. Based on our participants' views, we have developed four sub-themes on improving support for South Asian trauma survivors, specifically adult CSA survivors.These sub-themes are: 4.1 Cultural humility, 4.2 Raising awareness among South Asian individuals, 4.3 Increased diversity among service providers and 4.4 Targeted resources.

**Cultural humility.** In our study, mental health professionals and key stakeholders recognised and accommodated cultural nuances for their South Asian clients. For example, these service providers claimed that they were aware of the extra level of shame and self-blame experienced by their South Asian clients associated with family reputation. Consequently, our participants would focus on validating their clients' emotions. Further, our participants would be **open to the unknown** when discussing religion and spirituality.

According to our participants, in the absence of a specific approach designed and validated for the South Asian population, current evidence-based Western therapeutic models appeared to be effective with clients' needs being prioritised. Participants addressed that by acknowledging the unique cultural and social contexts of South Asian clients, service providers would be capable of making **appropriate adjustments** to their standard practices. Their practice included using language-appropriate communication if available and recognising specific cultural stigmas.

*No wonder it's so difficult to talk about these things if all your life you've been told it's, it's shameful to talk about them, it's dishonourable, it's disrespectful, that this even happened to you.*

*- Academic and Clinical psychologist, NHS*

**Raising awareness among South Asian community.** Our participants highlighted that some South Asian clients affected by trauma sometimes lacked mental health literacy, held misconceptions about CSA, and were unclear about mental health services. Our participants emphasised the need to raise awareness among the South Asian population in the UK. Participants suggested that this would involve education about the symptoms and impacts of mental health issues, the process and effect of psychological therapies or other treatments, as well as potentially distorted perceptions of CSA. As the wider community becomes informed about CSA and its impact, survivors could potentially have an increased likelihood of seeking help.

As mentioned by one mental health professional, some South Asian CSA survivors held strong beliefs that mental health problems, especially those related to their abusive experiences, were supposed to be processed alone, without seeking external support. Further, the professional highlighted that some survivors sought help when the impact of their abuse experiences on their mental health became unbearable. Mental health professionals in our study suggested that by improving mental health literacy, South Asian CSA survivors might be able to recognise signs of mental health struggles and abusive behaviours and understand the importance of professional support.

*A lot more awareness. A lot more, sort of local community centres where you can go and feel like you're part of the community that you trust […]*

*- Psychotherapist, National mental health association*

**Increased diversity among service providers.** Some participants explained that meeting clinicians or stakeholders who shared cultural identities with clients would help build a therapeutic relationship between them. For one of our participants, being non-British also helped build a connection with their South Asian clients.

*Some clients don't wanna talk to someone who's also South Asian, but majority of my clients love it when I say yeah, I get it. I went through that. I went through not being allowed to go out growing up. There's that real connection.*

*- Psychotherapist*

According to these mental health professionals and stakeholders, their South Asian survivor clients would often feel more comfortable when working with providers who shared similar cultural backgrounds or had a deep understanding of their cultural context. Therefore, an **increased representation of cultural identities** among service providers seemed to be helpful in building connections with clients. Mental health professionals noted that existing service providers with knowledge of various cultures would also help provide an inclusive environment.

*More people of diversity should be hired as clinical psychologists in general so that they are more representative when it comes to treating clients.*

*- Academic and Clinical psychologist, NHS*

**Targeted resources.** Mental health professionals and stakeholders in our study proposed the use of targeted resources that catered to the specific needs of South Asian clients affected by. Our participants suggested that drafting and explaining information sheets and other relevant **documents for clients in various languages** could encourage potential clients to seek help. In addition, they suggested that having resources in clients' native language could largely facilitate conversation and provide a sense of care. These professionals recommended that the content of psychoeducation could focus on issues that the South Asian population needed to know the most, including the necessity of professional intervention, confidentiality, and the process of treatment or support. Having some of the worries ruled out through psychoeducation could improve the engagement of their South Asian clients in psychosocial interventions.

*It would be great if we could offer services in different languages. And for the written resources we have on our website, very plain language English, […] It would be lovely to have that in a lot of different languages.*

*- Key stakeholder and Charity chair*

Stakeholders and mental health professionals in this study suggested **disseminating relevant information in venues and events relevant to South Asian culture.** They said that resources could be distributed in areas with higher concentrations of South Asian communities to enhance accessibility without the added burden of travelling long distances. A further benefit they said, would be the likely increase in the breadth of the community members receiving information on CSA and its impact. The participants suggested that this could help reduce stigma and misunderstanding about CSA, mental ill-health, and psychotherapy.

*We can go to the temples. We can go to weddings, you know, where you've got the bridal parties and everything before weddings, and we can talk to groups of women, and we can talk to groups of boys, and we can educate in those faiths schools about healthy sexual, and actually healthy relationships.*

*- Key stakeholder and Charity chair*

Mental health professionals in our study highlighted the accentuated need to **adapt Western-based interventions and assessments** for various cultures including South Asians. They indicated that current diagnostic scales may not accurately capture the experiences of South Asian clients. Professionals also suggested that additional consideration would be needed when reviewing assessment results and understanding the individual symptoms.

*I think that there's a huge gap at the moment, actually in terms of, adapting our interventions and our assessments for different populations […]*

*- Clinical psychologist, NHS*

## Discussion

Our study aimed to provide insights from mental health professionals and key stakeholders working with South Asian adult survivors of CSA in the UK. The findings suggest some preliminary insights into barriers that may affect South Asian CSA survivors' access to mental health support: pragmatic concerns, particularly financial and language barriers as well as culture-specific factors including shame, stigma, intense fear of community ostracisation and degradation of family honour. We conceptualised these barriers as having a bilateral relationship with the need to improve the informal and formal support for South Asian CSA survivors in the UK. Our study findings suggest that professionals perceive their South Asian survivor clients to rely on their families and community. They further reflected on the need to offer culturally tailored materials and ensure treatment by culturally sensitive professionals for this ethnic group.

A critical aspect that emerged from our study was the apparent lack of mental health literacy among the South Asian community. There is a lack of personalised resources for the specific demographic, given the "one-size-fits-all" model in the UK [36]. Our findings suggested the need to tailor psychoeducation for South Asian communities by providing resources in various languages and approaching the population through relevant and accessible venues. In addition to improving mental health literacy, this tailored approach could increase the likelihood of community engagement with mental health resources [37].

There is evidence to suggest that South Asian community members, as compared to British service users, have higher rates of mental ill-health, but lower rates of service utilisation [Gnanapragasam & Menon, 2021]. This aligns with our findings where professionals reported better awareness of mental health resources among non-South Asian clients in part attributable to relatively less stigma as compared to that in the South Asian community. Across interviews, professionals most frequently described their South Asian clients' symptoms consistent with PTSD/CPTSD, alongside depressive and anxiety symptoms; self-blame and low self-esteem were common. These patterns mirror international evidence on adult

CSA survivors in non-UK settings [38]. Further, our findings, in line with other studies, suggest that this ethnic group face a dual challenge of excessive stigma from within the community [23], and limited engagement with mental health services due to disbeliefs about mental health intervention [39]. Meanwhile, our study note that clients' religious and spiritual beliefs potentially influence their understanding and management of psychological distress. This observation aligns with existing literature indicating that spirituality and religious practices play a substantial role to mental health experiences in South Asian communities [14].

Our study suggested potentially a mismatch between South Asian cultural norms and those in the UK where they were seeking support. This cultural dichotomy could lead to conflicts in identity formation and expression [40]. In therapeutic settings, Western norms of openness and individuality may contradict South Asian norms of restraint and collectivism [41]. Such disparity might be represented in the contrasting expectation held by service users and service providers [42]. Some participants in our study reported that having personal experiences navigating between Western and South Asian cultures helped them build stronger therapeutic relationships with their South Asian clients who faced similar cultural challenges. Our findings highlighted the potential need for improving sensitivity towards such an experience of cultural mismatch. We propose the need for developing specialist training to work effectively with South Asian CSA survivors.

We found that for South Asian communities, informal support systems such as family networks could play a crucial role in mental health support. The strong familial ties and collectivist ideations within South Asian cultures could provide unique support that was more accessible and effective for individuals with mental health problems [43]. It could be beneficial due to the timeliness and cultural sensitivity of the support offered and could potentially minimise stigma against mental ill-health [44].

## Strengths and limitations

This study had several strengths. First, this is the first known study to explore professionals' views of the needs of South Asian CSA survivors in the UK, along with professionals' perspectives on the current treatment and support needs of this population. Second, the insights gained through this study underscore the need for culturally adapted mental health interventions that are accessible to the South Asian CSA survivors. Third, our findings could be valuable for mental health professionals and policymakers aiming to deliver services to culturally diverse populations. Fourth, our participants were from diverse ethnicities, providing a more comprehensive understanding of the subject. Finally, several mechanisms that this study identified, such as stigma, interpreter-mediated confidentiality, cultural framing of emotion, may generalise across other minority communities, provided that appropriate regulations and contextual considerations are taken into account [45].

This study also had some limitations that needed to be acknowledged. First, whilst the study included participants from diverse ethnic backgrounds, the study could have benefitted from more diversity in other participant characteristics. Most participants were female, and all were recruited in England and majorly practised in London. Insights from more male practitioners, as well as those practicing in other areas of the UK could be valuable. Second, we interviewed only two stakeholders, which might affect the comprehensiveness of the findings. While clinicians provided crucial insights into the therapeutic processes and client interactions, the perspectives of other mental health professionals, such as social workers, counsellors, and mental health nurses, as well as key stakeholders like community leaders, policymakers, and cultural mediators, were not included. Third, this study did not include the lived experiences of survivors. Given that many South Asian CSA survivors face significant obstacles to seeking support, professionals' accounts inevitably capture the limited challenges and needs that are visible to services, while more hidden lived experiences remain underrepresented. This is a crucial next step in advancing work in this area. The perspectives and experiences of survivors themselves would be crucial for a comprehensive understanding of the challenges they face and the support they most require. Although we conducted in-depth interviews, the recruited sample may not sufficiently represent the views of all UK professionals and their South Asian CSA survivors.

**Implications for policy, practice, and research**

The findings of this study underscore the importance of incorporating cultural competence into mental health services provided to South Asian CSA survivors in the UK. Policies could advocate for training on cultural humility and competence by implementing relevant training for healthcare providers, specifically addressing implicit biases and enhancing providers' skills in culturally sensitive communication. One finding from this study indicated that discouraging responses from service providers not only jeopardised the therapeutic relationship but also severely undermined service users' trust in the healthcare system. While individual biases might be unavoidable, it would be essential to develop anti-bias programs that increase professionals' awareness of them and minimise their impact, and incorporated such courses into the training procedure. In addition, service users could benefit if services put policies in place to recruit mental health professionals from diverse backgrounds, including those who share both Western and South Asian cultures, which could present as a bridge between the two cultures. As addressed by several participants from this study, there were some pragmatic concerns of service users that were rooted in language difficulties and financial concerns. Policy changes such as funding for language services or incorporation of multilingual digital platforms could be potentially beneficial. Further collaboration with local charities and religious organisations that were more accessible to South Asian service users could also be helpful.

By exploring the complex interplay between cultural identity, awareness, and the effectiveness of mental health intervention in diverse communities, this study aimed to advance the understanding of the unique challenges faced by South Asian trauma survivors living in the UK, contributing to the fields of mental health and multicultural counselling [46]. This study suggested potential changes in mental health care practices. As suggested by mental health professionals in our study, assessment scales should be used cautiously, as they might not fully capture the experiences of South Asian service users. A more thorough effort to understand the symptoms of service users would be necessary, which required comprehensive information collection and prompt communication. During the formulation process, it would be crucial to consider and integrate specific cultural factors such as cultural values, family dynamics and spiritual frameworks common within South Asian communities to better understand the needs of service users. Ideally, the development and validation of more culturally tailored interventions that reflected the unique experiences of South Asian CSA survivors living in the UK would be beneficial. Additionally, incorporating community feedback at each stage would help continuously refine these practices. In particular, this study underscores the need to adapt existing interventions to align more closely with South Asian CSA survivors' cultural contexts. These adaptations may involve using language-appropriate materials, incorporating psychoeducation plans that acknowledge cultural specific concepts, and delivering such plans in community venues, such as temples, gurdwaras, mosques, weddings, and other cultural events. Meanwhile, interventions could be delivered through trusted community organisations or in collaboration with religious leaders to increase engagement. Culturally adapted models should be co-developed with service users and community representatives to ensure they are both effective and accessible.

Employing semi-structured in-depth interviews, this study explored the nuanced perspectives of mental health professionals and key stakeholders, enhancing the depth of data on the intersection of cultural factors and mental health service utilisation. This study supplements previous exploration of complex themes such as stigma, decision-making, and help-seeking behaviours of South Asian communities through qualitative approaches [47]. Given the limitations of this current study, future research should aim to include a more balanced representation of mental health professionals and key stakeholders to provide a more holistic understanding of the challenges and opportunities in supporting South Asian CSA survivors. Furthermore, including the direct voices of survivors in research is imperative. Their firsthand accounts are crucial for providing a comprehensive view of their needs and the challenges they face, which ensures that support strategies are genuinely reflective of and responsive to the lived realities of South Asian CSA survivors. Future research should prioritize ethically sound approaches to include survivor perspectives, potentially through partnerships with community organisations, and using exploratory and mixed methods approaches.

## Conclusion

Our study, to the best of our knowledge, is the first in-depth exploration of UK mental health professionals and key stakeholders' views of the needs of and treatments and support available for South Asian trauma survivors, specifically adult CSA survivors. Overall, mental health professionals reported frequent observation of post-traumatic symptoms, depression, and anxiety, and highlighted barriers specific to the survivors' circumstances and those related to professionals. These barriers have a bilateral relationship with improving formal and informal sources of support for them in the UK. There is scope to adapt interventions for survivors in the South Asian community. This would require an exploration of South Asian CSA survivors' psychosocial needs and their views on the modifications needed to current services.

## Supporting information

**S1 File. Topic Guide for Interviews with Mental Health Professionals in UK.**
(PDF)

**S2 File. Topic Guide for Interviews with Key Stakeholders in the UK.**
(PDF)

## Acknowledgments

We would like to thank all the mental health professionals and stakeholders who participated in our study and shared their views and experiences with us.

## Author contributions

**Conceptualization:** Rebecca Appleton, Jo Billings, Shivangi Talwar.

**Data curation:** Yuqian Chen, Shivangi Talwar.

**Formal analysis:** Yuqian Chen, Rebecca Appleton, Jo Billings, Shivangi Talwar.

**Investigation:** Yuqian Chen, Shivangi Talwar.

**Methodology:** Rebecca Appleton, Jo Billings, Shivangi Talwar.

**Project administration:** Yuqian Chen, Eugenia Drini, Shivangi Talwar.

**Resources:** Yuqian Chen, Rebecca Appleton, Jo Billings, Shivangi Talwar.

**Software:** Yuqian Chen.

**Supervision:** Rebecca Appleton, Eugenia Drini, Jo Billings, Shivangi Talwar.

**Validation:** Yuqian Chen, Shivangi Talwar.

**Visualization:** Yuqian Chen, Shivangi Talwar.

**Writing – original draft:** Yuqian Chen.

**Writing – review & editing:** Yuqian Chen, Rebecca Appleton, Eugenia Drini, Jo Billings, Shivangi Talwar.

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
