## [Decision Letter · Decision Letter 0]

24 Apr 2025

PMEN-D-25-00080

Needs of South Asian Adult Survivors of Childhood Sexual Abuse in the UK: A Qualitative Study with UK Mental Health Professionals and Key Stakeholders

PLOS Mental Health

**Dear Dr. Talwar,**

Thank you for submitting your manuscript to PLOS Mental Health. After careful consideration, we feel that it has merit but does not fully meet PLOS Mental Health’s publication criteria as it currently stands. Therefore, we invite you to submit a revised version of the manuscript that addresses the points raised during the review process.

EDITOR:

Address the comments raised by the reviewers

We look forward to receiving your revised manuscript.

Kind regards,

Kizito Omona, PhD

Academic Editor

PLOS Mental Health

Journal Requirements:

Additional Editor Comments (if provided):

Address the comments of the reviewers

Reviewers' comments:

Reviewer's Responses to Questions

**Comments to the Author**

1. Does this manuscript meet PLOS Mental Health’s publication criteria ? Is the manuscript technically sound, and do the data support the conclusions? The manuscript must describe methodologically and ethically rigorous research with conclusions that are appropriately drawn based on the data presented.

Reviewer #1: Yes

Reviewer #2: Yes

2. Has the statistical analysis been performed appropriately and rigorously?

Reviewer #1: Yes

Reviewer #2: Yes

3. Have the authors made all data underlying the findings in their manuscript fully available (please refer to the Data Availability Statement at the start of the manuscript PDF file)?

Reviewer #1: Yes

Reviewer #2: No

4. Is the manuscript presented in an intelligible fashion and written in standard English?

Reviewer #1: Yes

Reviewer #2: Yes

5. Review Comments to the Author

Reviewer #1: I'm glad to have had the opportunity to review this paper.

The paper explores the barriers and support needs of South Asian adult survivors of childhood sexual abuse in the UK from the perspectives of mental health professionals and key stakeholders. The study is well-structured, with clear findings highlighting the cultural and systemic challenges in mental health service access. Thematic structure is logical, making the findings easy to follow.

However, while the study provides valuable insights, its sample size, lack of survivor perspectives, and limited geographic scope are significant limitations. Nevertheless, the study includes a mix of ethnic backgrounds and professional roles, strengthening its credibility.

The manuscript can be of interest upon making the following revisions.

1. Within the introduction, you may incorporate a direct comparison to prior studies on CSA survivors of other ethnic groups to highlight what makes South Asian experiences unique.

2. The introduction mentions barriers to care but does not contextualize them within existing models of cultural competency.

3. The rationale for qualitative methods is briefly stated but not justified. The authors may expand on why Reflexive Thematic Analysis was the most appropriate choice compared to other qualitative methods (e.g., Interpretative Phenomenological Analysis).

4. The purposive and snowball sampling approach is mentioned, but it is unclear how participants were screened and selected. Authors can specify inclusion/exclusion criteria (e.g., minimum years of experience, type of professional role). Furthermore, the study only includes professionals and stakeholders, with no direct survivor interviews. Authors can provide a stronger justification for this decision and discuss its implications in the limitations.

5. The paper calls for culturally adapted interventions but lacks specific action steps. Clearer and more specific implementation strategies should be provided.

6. Please clearly state that future studies should incorporate survivor narratives to gain first-hand insights. A future mixed-methods study incorporating survivor interviews would enhance depth and applicability.

7. A thorough proofreading pass is needed before submission to check for grammatical errors.

Reviewer #2: Thank you for the opportunity to read such an impactful topic. I have some minor recommendations for you to address, kindly go through them:

Abstract: The abstract looks fine but I would recommend you to remove the subheadings (background and aims, method, findings, conclusion) from the abstract and make a single paragraph and concise the abstract a little more because it is exceeding the word count limit of 300 words. Check for spelling errors in findings subheadings: "Help from families and witin (within) the community as well as ...." , " This may invovle (involve) improving the ..."

Introduction and Discussion: Dear authors, you have wonderfully identified several barriers to access mental health services in South Asian countries in both Introduction and Discussion sections, especially for the victims of CSA, but I would encourage you to explore a little more on it. For example, one of the common barriers in of accessing Mental Health Care in South Asia is belief in superstition and easy accessibility of traditional/religious healers, village doctors/pharmacy shopkeepers. I am adding a reference for your convenience (https://doi.org/10.1007/s10597-021-00790-0) but you can explore more on this. It will help you to build strong argument, comparison and contrast in the discussion part particularly.

Again, the introduction is too long, try to concise it but, consider adding more explicit gaps in the existing literature, like, while you mention the lack of evidence, elaborating on why this gap is particularly critical for the South Asian community could strengthen the argument for the study’s significance.

Spelling Errors (introduction) 1st paragraph: "One of the most prevalent and debilating (debilitating) developmental...", 3rd paragraph: " Adult CSA survivors of South Asian origin often report further specific barriers in accesing (accessing) support...", 4th paragraph: "Alongside the NHS, numerous charities and organisations in the UK offer hlplines (helplines)...", 5th paragraph: "accessing community support networks, practicing cultual (cultural) healing modalities, engaging in social advocacy, and developing intentional self-care practies (practices). Studies also call for clutually-(culturally) informed interventions...", 6th paragraph, "demonstrated that evidence was availvale (available) from...", "unpleasant repsponses (responses) to...", "evidence base for specific treaments (treatments), with...", 8th paragraph: "with Saouth (South) Asian adult CSA...", "work in the USA, a crosssectional (cross-sectional) survey...", last paragraph: "UK mental health profesionnals (professionals)..."

(discussion) 1st paragraph: "atment by cultually (culturally) sensitive, "limited engagment (engagement) with.."

I have gone through the rest of the manuscript where I don't have any specific recommendations. The findings are appropriate with rigorous thematic analysis. I understand the time, hard work and effort while writing a qualitative research paper, still I would highly encourage you to recheck manuscript for spelling errors too.

Good luck with the submission.

6. PLOS authors have the option to publish the peer review history of their article (what does this mean? ). If published, this will include your full peer review and any attached files.

**Do you want your identity to be public for this peer review?** For information about this choice, including consent withdrawal, please see our Privacy Policy .

Reviewer #1: **Yes: ** Sophia Ahmed

Reviewer #2: **Yes: ** Rehnuma Abdullah

---

## [Decision Letter · Decision Letter 1]

7 Aug 2025

PMEN-D-25-00080R1

Needs of South Asian Adult Survivors of Childhood Sexual Abuse in the UK: A Qualitative Study with UK Mental Health Professionals and Key Stakeholders

PLOS Mental Health

Dear Dr. Talwar,

Thank you for submitting your manuscript to PLOS Mental Health. After careful consideration, we feel that it has merit but does not fully meet PLOS Mental Health’s publication criteria as it currently stands. Therefore, we invite you to submit a revised version of the manuscript that addresses the points raised during the review process.

EDITOR:

I have seen very interesting comments from two additional reviewers, which I believe can make your work better before we accept to publishDo not make major revision that alters the already better part of this study.

Please submit your revised manuscript by **6th September 2025. ** If you will need more time than this to complete your revisions, please reply to this message or contact the journal office at mentalhealth@plos.org. Please include the following items when submitting your revised manuscript:

We look forward to receiving your revised manuscript.

Kind regards,

Kizito Omona, PhD

Academic Editor

PLOS Mental Health

Journal Requirements:

Additional Editor Comments (if provided):

Reviewers' comments:

Reviewer's Responses to Questions

**Comments to the Author**

1. If the authors have adequately addressed your comments raised in a previous round of review and you feel that this manuscript is now acceptable for publication, you may indicate that here to bypass the “Comments to the Author” section, enter your conflict of interest statement in the “Confidential to Editor” section, and submit your "Accept" recommendation.

Reviewer #3: All comments have been addressed

Reviewer #4: All comments have been addressed

Reviewer #5: (No Response)

Reviewer #6: All comments have been addressed

2. Does this manuscript meet PLOS Mental Health’s publication criteria ? Is the manuscript technically sound, and do the data support the conclusions? The manuscript must describe methodologically and ethically rigorous research with conclusions that are appropriately drawn based on the data presented.

Reviewer #3: Yes

Reviewer #4: Yes

Reviewer #5: Yes

Reviewer #6: Partly

3. Has the statistical analysis been performed appropriately and rigorously?

Reviewer #3: N/A

Reviewer #4: (No Response)

Reviewer #5: N/A

Reviewer #6: N/A

4. Have the authors made all data underlying the findings in their manuscript fully available (please refer to the Data Availability Statement at the start of the manuscript PDF file)?

Reviewer #3: Yes

Reviewer #4: Yes

Reviewer #5: No

Reviewer #6: Yes

5. Is the manuscript presented in an intelligible fashion and written in standard English?

Reviewer #3: Yes

Reviewer #4: Yes

Reviewer #5: Yes

Reviewer #6: Yes

6. Review Comments to the Author

Reviewer #3: The author has adequately responded to the reviewers' comments.

Reviewer #4: Strengths:

1. Relevance and Novelty: The study tackles an important gap in the literature by focusing on the intersection of cultural identity and CSA survivorship, particularly within the South Asian community in the UK. This is a timely and valuable contribution to mental health research.

2. Qualitative Approach: The use of qualitative methods, including interviews with mental health professionals and stakeholders, is well-suited to capturing nuanced perspectives on a sensitive topic.

3. Cultural Sensitivity: The manuscript demonstrates an awareness of cultural factors influencing survivors’ experiences, which is critical for informing culturally competent mental health services.

Areas for Improvement:

1. Clarity of Research Aims:

- The introduction could more explicitly state the research questions or objectives. While the focus on needs is clear, specifying whether the study aims to explore barriers, service gaps, or specific support needs (or all three) would enhance clarity.

- Consider refining the title to reflect the study’s scope more precisely, e.g., by specifying whether the focus is on service provision, cultural barriers, or survivor experiences as perceived by professionals.

2. Methodology:

- Participant Selection: Provide more detail on the recruitment process and inclusion criteria for mental health professionals and stakeholders. For instance, how were “key stakeholders” defined, and what expertise did they bring? This would strengthen the study’s credibility.

- Sample Size and Diversity: The manuscript would benefit from justifying the sample size and discussing the diversity of participants (e.g., professional roles, cultural backgrounds). If the sample was predominantly non-South Asian, address how this might influence perspectives on South Asian survivors’ needs.

- Data Analysis: The description of the thematic analysis process is somewhat brief. Elaborate on the coding framework, whether it was inductive or deductive, and how themes were derived. Mentioning inter-rater reliability or member-checking (if applicable) would enhance methodological rigor.

3. Findings:

- The themes presented are compelling but could be better supported with direct quotes to illustrate professionals’ and stakeholders’ perspectives. This would add depth and authenticity to the findings.

- Some findings (e.g., cultural stigma, lack of tailored services) align with existing literature. Clarify how your study extends or diverges from prior research to highlight its unique contribution.

- Consider organizing the findings section to align more clearly with the research aims, ensuring each theme directly addresses the needs of South Asian CSA survivors.

4. Discussion and Implications:

- The discussion could further explore the practical implications of your findings for mental health service providers. For example, what specific training or resources could address identified gaps? Providing actionable recommendations would enhance the study’s impact.

- Address potential limitations more thoroughly, such as the reliance on professionals’ perspectives rather than survivors themselves. Discuss how this choice shapes the findings and what future research could complement this approach (e.g., studies directly involving survivors).

- The manuscript could benefit from a brief discussion of how findings might apply to other minority ethnic groups or contexts, broadening its relevance.

5. Ethical Considerations:

- While ethical approval is mentioned, expand on how sensitive topics were handled during interviews (e.g., ensuring participant wellbeing, managing potential distress). This is particularly important given the study’s focus on a vulnerable population.

6. Clarity and Structure:

- Some sections, particularly the results, could be more concisely written to avoid repetition. For example, streamline overlapping themes or sub-themes to improve readability.

- Ensure consistent terminology (e.g., “South Asian survivors” vs. “adult survivors of CSA”) throughout the manuscript for clarity.

- A table summarizing key themes and sub-themes could help readers quickly grasp the study’s findings.

Additional Suggestions:

- Consider integrating a brief literature review on existing studies about CSA survivors in minority communities to contextualize your findings within a broader framework.

- If possible, include a section on how the findings were disseminated or could be shared with relevant stakeholders (e.g., mental health organizations, South Asian community groups) to maximize real-world impact.

Reviewer #5: General comments

It is an interesting, unique study giving insight into the “Needs of South Asian Adult Survivors of Childhood Sexual Abuse in the UK: A Qualitative Study with UK Mental Health Professionals and Key Stakeholders.” However, here are some comments for your attention;

Abstract

The abstract is clear, however;

Is it possible to adapt the aim of the study that is captured in the last paragraph of the introduction, “to explore the perspectives of UK mental health professionals and key stakeholders on the treatment and support needs of South Asian adult CSA survivors,” for uniformity purposes?

Introduction

The introduction lacks a clear definition of childhood sexual abuse (CSA), including how CSA presents in the survivors.

Could you include some statistics (prevalence) on childhood sexual abuse (CSA) survivors in the UK?

Methods

Under the sub-heading of data collection, use “interview guide” instead of “topic guide.”

Was the interview guide pretested after its development, and where was it pretested?

How did you come up with a sample size of five mental health workers and two stakeholders? Was this it after reaching redundancy?

Results

Is it possible to include more details in “Table 1: Characteristics of participants,” such as duration of practice, as this may go hand in hand with experience as regards offering treatment to CSA survivors?

Discussion

In the first paragraph of the discussion, the statement “We also found that there may be a need to better understand the needs of South Asian CSA survivors in the UK” is confusing, as the whole study is about this. I suggest that this can be removed.

Regarding reflexivity, the authors elaborated it in the methodology section; however, during the discussion of the themes (findings), they did not acknowledge/incorporate their positionality in the narration of the themes. A reflexivity statement is key at the end of each of the broad themes discussed.

Reviewer #6: General comments and some key concerns:

1. The manuscript is addressing an important topic on the “Needs of South Asian Adult Survivors of Childhood Sexual Abuse in the UK: A Qualitative Study with UK Mental Health Professionals and Key Stakeholders”. The barriers and support needs of South Asian adult survivors of childhood sexual abuse in the UK is critical in the perspectives of the of mental health professionals and key stakeholders. However, there some comments that needs to be addressed including the following:

Language

There is excessive use of first and second person in sentence construction and suggest that they are reduced in the abstract and all through the manuscript. They should be used only where necessary.

Note: The authors need to align all the components of the study on the perspectives of the UK Mental Health Professionals and Key Stakeholders on the needs of South Asian Adult Survivors of Childhood Sexual Abuse in the UK since the South Asian Adult Survivors of CSA did not participate in the study.

Title

The authors need to paraphrase the title to reflect the actual study participants in the study since the actual South Asian Adult Survivors of Childhood Sexual Abuse (CSA) did not participant in the study. Suggestion “Needs of South Asian Adult Survivors of Childhood Sexual Abuse in the UK: Perspective from UK Mental Health Professionals and Key Stakeholders”. The title and the objective need to be paraphrased to specifically reflect on the actual study participants.

2. Abstract

The first sentence on Childhood sexual abuse (CSA), the authors should provide information where this problem is as a background to the study; and suggest UK where the study was conducted (see the title). The authors need to address the Childhood sexual abuse (CSA) globally and then narrow down to the UK in both the abstract and introduction. The aim of the study in the abstract is not in line with the title and therefore it needs to be paraphrased. The authors should clearly explain who are the 7 study participants and 2 stakeholders. Were the adult survivors of CSA of South Asian origin, the study participants? It is not clear. Was there any tool that was used in the interviews? The authors should provide the key finding from the study and the conclusion should be based on those findings. The authors need to paraphrase the objective to specifically reflect on the actual study participants in the study.

3. Introduction

The authors should stick to childhood sexual abuse (CSA) and not global trauma, and this should highlight the CSA problem globally and then narrow down to the South Asian Adult Survivors of Childhood Sexual Abuse in the UK and the development of mental illnesses especially the Post-Traumatic Stress Disorder (PTSD). The authors should also explain the need of the diverse CSA globally both developed and under-developed including the South Asia that have been reported which possibly may differ from those of the South Asian origin i.e. do CSA in South Asia, sub- Saharan Africa and western Europe and many others differ or share similarity in this aspect. This should be highlighted in a holistic way and not only the South Asia since they have different cultural diversity and resilience. The authors should clearly explain the knowledge gap the study tried to address. Then the aim of the study should be in line with the title.

4. Methods section

The authors should replace “Methods” with “Methodology”. The authors should clearly state the study setting and the study participants. Who were the actual adult CSA survivors? The authors should also explain why the CSA were not included in the study so that their needs perspectives are known other than getting them from the professionals. For the key stakeholders, they wouldn’t be individuals with experience supporting South Asian CSA survivors; they should be the actual CSA survivors. The authors should explain how snowball sampling was conducted and who was used to identify those individuals of interest. The sample size seems to be small for the generalization of the finding and can the authors explain why this was so? Was there a tool that was used as a guide during the intervention? For how long were the interviews for each study participants and when were they conducted? This information need to be incorporated in each relevant sub-section. On the subsection of quality and rigor, the authors recruited participants from diverse backgrounds, including ethnicity, gender, age, and occupation, to capture a range of opinions, these should be clearly explained. And given this criteria, how many participants represented each of these criterion (takes back to the sample size and generalization of the findings). On the Reflexivity, the authors mention that all team members, how many teams, and the numbers in each team or individuals that were making each team.

5. Results

The study explored the needs of South Asian Adult Survivors of Childhood Sexual Abuse in the UK, the lack of the representation of South Asian Adult Survivors of CSA as the title and aim suggests limits the key findings on this study group. Furthermore, only 2 British Indian (Table 1) who were not CSA represented the CSA group with lack of that experience is limitive for the generalization of the CSA experiences. The title and the objective need to be paraphrased to specifically reflect on the actual study participants. The findings should be presented as % (n) in table 1 and to one decimal place. The mean age should have a standard deviation. In table 2, the finding on the challenges faced by survivors with professionals should be written as “challenges faced by survivors reported by professionals” since the survivors did not participate in the study. The authors should explain the common mental disorders encountered by the South Asian Adult Survivors of Childhood Sexual Abuse in the UK that is reported and this should also appear in the discussion.

6. Discussion

The authors should explain the common mental disorders encountered by the South Asian Adult Survivors of Childhood Sexual Abuse in the UK that is reported and how this is related previous studies elsewhere. Among the limitations of the study mentioned, the small sample size on each criterion including the diverse backgrounds, including ethnicity, gender, age, and occupation, to capture a range of opinions should be included. On the implications for policy, the challenge of the small sample size and the lack of the lived examples in the study limit the policy development and the intervention development towards the South Asian Adult Survivors of Childhood Sexual Abuse in the UK since findings are based on the individual opinions of the healthcare professionals.

7. Conclusion

The authors should clearly provide the key finding from the study especially on the common mental disorders encountered by the CSA as reported by the health professionals and the key challenges encountered by the CSA in UK.

7. PLOS authors have the option to publish the peer review history of their article (what does this mean? ). If published, this will include your full peer review and any attached files.

**Do you want your identity to be public for this peer review?** For information about this choice, including consent withdrawal, please see our Privacy Policy .

Reviewer #3: No

Reviewer #4: **Yes: ** Prof. Dr. Veysi ÇERİ

Reviewer #5: No

Reviewer #6: No

---

## [Editor Report · Decision Letter 2]

19 Sep 2025

Support Needs of South Asian Adult Survivors of Childhood Sexual Abuse in the UK: Perspectives of UK Mental Health Professionals and Key Stakeholders

PMEN-D-25-00080R2

Dear Dr. Talwar,

We are pleased to inform you that your manuscript 'Support Needs of South Asian Adult Survivors of Childhood Sexual Abuse in the UK: Perspectives of UK Mental Health Professionals and Key Stakeholders' has been provisionally accepted for publication in PLOS Mental Health.

Best regards,

Kizito Omona, PhD

Academic Editor

PLOS Mental Health

Thank you for revising your paper based on reviewer comments